

# A comparison of Loon balloon observations and stratospheric reanalysis products

Leon S. Friedrich[1], Adrian J. McDonald[1], Gregory E. Bodeker[2], Kathy E. Cooper[3], Jared Lewis[2], and Alexander J. Paterson[1]

[1]Department of Physics and Astronomy, University of Canterbury, New Zealand.
[2]Bodeker Scientific, Alexandra, New Zealand.
[3]X Project Loon, 1600 Amphitheatre Parkway, Mountain View, CA 94043.

*Correspondence to:* Adrian McDonald
(adrian.mcdonald@canterbury.ac.nz)

**Abstract.** Location information from long duration super pressure balloons flying in the Southern Hemisphere lower stratosphere during 2014 as part of X Project Loon are used to assess the quality of a number of different reanalyses including National Centers for Environmental Prediction Climate Forecast System version 2 (NCEP-CFSv2), European Centre for Medium-Range Weather Forecasts (ERA-Interim), NASA Modern Era Retrospective-Analysis for Research and Applications (MERRA), and the recently released MERRA version 2. Balloon GPS location information is used to derive wind speeds which are then compared with values from the reanalyses interpolated to the balloon times and locations. All reanalysis data sets accurately describe the winds, with biases in zonal winds of less than $0.37\mathrm{m/s}$ and meridional biases of less than $0.08\mathrm{m/s}$. The standard deviation on the differences between Loon and reanalyses zonal winds is latitude dependent, ranging between 2.5 and 3.5 $\mathrm{m/s}$ increasing equatorward.

Comparisons between Loon trajectories and those calculated by applying a trajectory model to reanalyses wind fields show that MERRA-2 wind fields result in the most accurate simulated trajectories with a mean 5 day balloon–reanalysis trajectory separation of $621\mathrm{km}$ and median separation of $324\mathrm{km}$ showing significant improvements over MERRA version 1 and slightly outperforming ERA-Interim. The latitudinal structure of the trajectory statistics for all reanalyses displays marginally lower mean separations between $15°\mathrm{S}$ and $35°\mathrm{S}$ than between $35°\mathrm{S}$ and $55°\mathrm{S}$, despite standard deviations in the wind differences increasing toward the equator. This is shown to be related to the distance travelled by the balloon playing a role in the separation statistics.

## 1 Introduction

X (an Alphabet Company, formerly known as Google[x]) Project Loon, hereafter referred to as Loon, aims to provide worldwide Internet coverage using a network of long-duration super-pressure balloons. These balloons fly in the stratosphere at approximately 20km altitude with flight durations averaging 55 days (maximum 187 days, median 42 days). In this study zonal and meridional wind speeds, derived from Loon location information obtained from the on-board GPS, are compared with interpolated winds from four different reanalyses. The reanalyses used are the European Centre for Medium-Range Weather



Forecasts (ECMWF) ERA-Interim reanalysis (Dee et al., 2011), NASA's Modern-era Retrospective Analysis for Research and Applications (MERRA) (Rienecker et al., 2011), the recently released MERRA-2, and the National Centers for Environmental Prediction (NCEP) Climate Forecast System Version 2 (CFSv2) analysis (Saha et al., 2011) (which we refer to as one of the reanalyses). The reanalyses assimilate a range of data to tightly constrain a global atmosphere-ocean climate model simula-

5 tion. Using satellite data, in-situ observations from radiosondes and other data sources, the reanalyses generate a data set that provides a best estimate of the state of the global atmosphere.

These reanalyses are often used to study stratospheric dynamical processes. In particular, reanalyses winds are used to compute forward and backward trajectories to trace the motion of air parcels. For example, a Lagrangian chemical box model can be used to determine ozone loss rates in an air parcel by measuring the concentration of ozone at various times while

keeping track of the parcel through isentropic trajectory modelling (Vondergathen et al., 1995). Trajectory analyses are also important for quantifying mixing between different air masses which can affect atmospheric chemistry. This is important as many chemical processes depend non-linearly on the concentrations of the reactants (Stohl et al., 2004) e.g. the rate of ozone loss in the stratospheric polar vortex (Tuck et al., 2003). Calculated trajectories are also used to infer various metrics of mixing (Nakamura, 1996; Haynes and Shuckburgh, 2000; Smith and McDonald, 2014). Determining trajectories is also

central to domain-filling techniques which allow fine scale structure in chemical constituent fields to be derived from space-based measurements (Sutton et al., 1994; Smith and McDonald, 2014). Loon flights are therefore also used to examine the accuracy of trajectories derived from the reanalyses.

Stohl et al. (2004) discuss the importance of reanalysis quality in mixing studies. In particular, features such as the polar vortex, which act as barriers to mixing, may be displaced in an analysis relative to the position a forecast from the previous

analysis would have predicted. The reason for such a displacement is unphysical and arises from the assimilation of observations. In a transport model used with these analyses, an air parcel may therefore find itself on the other side of a mixing barrier without actually crossing it in a physically meaningful way. Thus, understanding the quality of the reanalyses fields is important in stratospheric chemistry studies.

Measurements of the stratospheric wind field are sparse. While routine radiosonde flights are made once, twice or four

25 times daily at more than 100 upper-air sites within the Global Observing System, because the resultant data are assimilated into the reanalyses, they cannot provide an independent verification of the quality of the reanalyses. Independent data from long duration balloon flights therefore provides a valuable assessment of reanalysis accuracy. The balloon–reanalysis comparison reported on here adds to the body of knowledge encompassed in previous studies, which used a range of models and balloon flights (Knudsen et al., 2002; Hertzog et al., 2004; Knudsen et al., 2006; Hertzog et al., 2006; Parrondo et al., 2007;

Boccara et al., 2008; McDonald and Hertzog, 2008; de la Camara et al., 2010; Podglajen et al., 2014). These previous studies have been performed in varied geographical regions, generally using fewer balloons than are used in the analyses reported here. To provide a context for the results reported on below, a brief summary of the key results from previous comparison studies is provided.

Hertzog et al. (2004) used six super–pressure balloons launched from high Northern latitudes to assess the quality of

35 ECMWF and NCEP/NCAR reanalyses in the lower stratosphere. The NCEP/NCAR reanalysis temperatures showed a 0.8 K





warm bias relative to the observations, while the ECMWF analyses showed a 0.3 K cold bias. The temperature observations exhibited small-scale fluctuations which Hertzog et al. (2004) attributed to mesoscale inertia-gravity waves. Both analyses accurately represented the winds with biases of less than 0.3 m/s and standard deviations ranging from 2.3 to 2.7 m/s. Trajectory comparisons suggested that ECMWF derived trajectories were more accurate than those determined using NCEP/NCAR wind fields, with trajectory errors after 15 days of 1000±1200 km for ECMWF and 2300±1300 km for NCEP/NCAR trajectories.

Knudsen et al. (2006) examined data from 11 balloons launched from Brazil in 2004. Relative to the balloon-based temperatures, the temperature extracted from the ECMWF operational analyses had a mean 0.9 K cold bias, with a standard deviation of 1.3 K. ECMWF winds showed biases of less than 0.4 m/s, with standard deviations of about 3 m/s, resulting in average trajectory separations of about 500 km after 5 days.

Hertzog et al. (2006) assessed the ECMWF ERA-40 and NCEP/NCAR NN50 reanalyses in the Southern Hemisphere upper troposphere and lower stratosphere based on comparisons with 480 super pressure balloon flights, most lasting longer than 100 days, from the 1971-72 Eole experiment. These comparisons indicated that, in the sub-polar latitudes, both NN50 and ERA-40 exhibited a cold bias of 3 K and 0.5 K respectively, while both had a warm bias of ~1 K in the tropics. The winds were found to have biases of ± 1 m/s, with latitude binned standard deviations ranging from 5 to 15 m/s.

Boccara et al. (2008) used data from 27 super pressure balloon flights, launched as part of the 2005 Antarctic Vorcore campaign, to examine the quality of ECMWF operational analysis and NCEP-NCAR NN50 reanalysis. The NN50 reanalysis showed a 1.51 K warm bias while the ECMWF analyses showed a 0.42 K cold bias. The winds in both reanalyses winds showed biases of less than 0.15 m/s, with standard deviations ranging between 2.4 and 3.4 m/s, with ECMWF performing better than NN50. These results indicated an improvement relative to those in Hertzog et al. (2006) which is likely related to the lack of data assimilated in the Southern Hemisphere prior to the satellite period. Boccara et al. (2008) attributed the small scale fluctuations in the wind and temperature data to gravity waves that were unresolved in the reanalyses. By applying a low-pass filter to remove these small scale fluctuations, they determined that a significant proportion of the standard deviation was a result of these perturbations. Trajectory separations were found to exceed 1000± 700 km after 5 days using NN50, and 10 days for ECMWF.

McDonald and Hertzog (2008) compared temperature measurements in the Antarctic stratosphere made by the CHAMP radio occultation satellite and in situ temperature measurements from Vorcore campaign balloons. The analysis compared near-simultaneous and co-located temperature observations made by these instruments and found excellent agreement between the temperatures measured in two very different ways. The mean bias between the data sets was -0.52 K, with CHAMP temperatures being cooler than the balloon-based measurements, with a standard deviation in the differences of 1.6 K. This paired data set also enabled McDonald and Hertzog (2008) to show that an empirical correction used to remove the influence of radiative heating on the balloon temperature sensors, a variant of which is commonly used to correct balloon-based temperature measurements, did not produce any additional bias.

Podglajen et al. (2014) used data from 3 equatorial long-duration balloon flights, launched in 2010, to examine the performance of ERA-Interim, MERRA, and ECMWF operational analysis. The results of the temperature comparisons were relatively similar to those of previous comparisons, with small warm biases (up to 1 K for MERRA), and standard deviations



ranging from 1.5 K for ECMWF to 2.2 K for MERRA. The analyzed winds, however, were found to show higher biases than similar analyses in the extra–tropics, with concomitant large differences in derived trajectories. All of the reanalyses were found to have zonal wind biases greater than 2 m/s, with the standard deviation of the reanalysis wind differences ranging from 3.5 m/s to 5.8 m/s. Detailed analysis of cases of persistent (more than 10 days) significant biases in the reanalyses,

with zonal wind biases and standard deviations of ∼9 m/s, suggested that these events corresponded to large-scale equatorial Kelvin and Yanai wave packets with small vertical wavelengths which were not resolved in the reanalyses. Podglajen et al. (2014) also discussed the likely causes of the poor representation of stratospheric equatorial waves and concluded that one of the key factors was the lack of wind speed observations assimilated by the analyses, particularly over the data-sparse eastern Pacific and Indian Ocean.

## 2 Data and Methodology

### 2.1 Balloon Dataset

In this study, 70 individual Loon balloon flights are examined, with flight durations varying from a few days to nearly 200 days. The flights occur predominantly in the Southern Hemisphere mid-latitudes, with the majority of the balloons being launched from Tekapo in New Zealand. The flight data occur over the period March 2014 to January 2015. Flight distribution

information is shown in Fig. 1. The pressure levels of the balloon flights vary between 30 and 70 hPa with an actively controlled altitude, although this active control is used relatively rarely, typically with multiple days between altitude changes. The Loon group use forecasts from the NCEP global forecast system (GFS), as well as forecasts from other sources, to simulate expected balloon trajectories. Based on these forecasts, decision are made by the Loon team to occasionally adjust the balloons' altitudes which is done by pumping air into or out of an internal bladder to modify the balloon density. While, super-pressure balloons

typically move along isopycnic (constant density) surfaces, during the rare occasions of altitude control, this is no longer the case. Intervals during which the altitude of a balloon is being modified can be clearly identified by very rapid changes in the pressure. In the following analysis, whenever a pressure change greater than 5 hPa occurs within one hour, the balloons is considered to be undergoing an altitude control manoeuvre and the data from that period are excluded from the subsequent analysis.

Each balloon data set includes three dimensional GPS position, pressure, and balloon lift-gas temperature, all of which are typically recorded at 1 minute intervals with occasional gaps due to telemetry failures. Although no specific details of the instruments used on each of the balloon flights is recorded, the Loon team have provided an upper bound on the uncertainties of the sensors viz. 1.5 hPa for pressure, 10 m for GPS location, and 10 K for temperature. The GPS uncertainty suggests an upper bound of 0.33 m/s uncertainty on derived wind speed measurements. The upper bound on the pressure sensor uncertainty

is rather large and could potentially lead to uncertainties when vertically interpolating the reanalyses data sets to the balloon locations.

Comparisons of Loon pressure sensor measurements with pressures extracted from reanalyses, where the reanalyses geopotential heights have been converted to geometric heights to allow direct comparisons with the GPS-referenced Loon data, indi-





cate that each individual balloon flight exhibits pressure sensor biases ranging from -0.5 to +1.70 hPa, in agreement with the provided uncertainty estimate. Mean biases against NCEP-CFSv2 reanalyses (Loon minus reanalyses) are $0.535\pm0.537$ hPa. Adjusting the pressure data for these biases has only minor impacts on the subsequent analysis. The temperature measurements, being a measure of the lift gas and not the ambient air, are of questionable scientific utility in the current context; their usability

is further examined in section 3.3.

## 2.2 Methodology

For the comparisons between the Loon observations and the reanalyses products a methodology very similar to that used in Boccara et al. (2008) is used to interpolate the reanalysis data to the temporal and spatial position of the balloon. A summary of the resolutions of the reanalysis products used in this study is provided in Table 1. Our interpolation scheme is a cubic spline

fit over 6 data points in both horizontal directions, log-pressure, and time. Simple bilinear interpolation schemes occasionally displayed signs of discontinuities in the reanalysis fields, likely related to the assimilation of data, which subsequently produced dynamical inconsistencies as previously identified in Stohl et al. (2004). The latitude and longitude GPS location data are combined with a simple finite difference calculation to derive the zonal and meridional winds which advect the balloons. Use of a five point derivative calculation scheme, which is more robust in the presence of noise, produces almost no difference in

the velocities derived, but is impacted more by occasional data gaps than the simple scheme, and was therefore not used in this study.

   A Lagrangian trajectory model was also used to compare trajectories derived from reanalyses against the balloon trajectories. Every six hours along a balloon flight, an 8-day trajectory was initialized. While super pressure balloons closely follow isopycnic surfaces, and hence isopycnic trajectories are generally used (Hertzog et al., 2004; Boccara et al., 2008; Podglajen et al.,

2014), in the model used here the vertical motion is also accounted for by setting the altitude of the modelled trajectory to correspond to the pressure level of the balloon, as is done by Knudsen et al. (2006). While this approach decreases the impact of potentially failing to recognize small altitude modifications, the range of potential trajectories is still limited by the occasional large altitude changes. Even when calculating trajectories with altitudes prescribed from the balloons, non-isopycnic altitude changes can exacerbate small separations in modelled and actual trajectories. Therefore, for the purposes of this analysis, any

trajectories that encounter non-isopycnic balloon altitude changes are truncated such that the data after the altitude shift are excluded from later analysis.

   The Lagrangian trajectory model used in this study was developed at the University of Canterbury and is a modified version of that used and discussed in Alexander et al. (2013), McDonald and Smith (2013) and Smith and McDonald (2014). It uses a fourth order Runge-Kutta algorithm, with a 10 minute time-step, with reanalysis wind speeds determined at the trajec-

tory position using the spatial-temporal interpolation scheme detailed above. A polar stereographic coordinate system is used equatorwards of 70 degrees to avoid the singularity at the pole.





## 3 Results

### 3.1 Winds

A sample of the zonal and meridional winds derived from one of the Loon GPS data sets, along with the corresponding
reanalysis winds, is shown in Figure 2. This flight is shown as an example since it exhibits a wide range of zonal wind
velocities. The comparison shows a good correspondence between the Loon observations and all four of the corresponding
reanalyses wind time series. While some differences are observed between the reanalysis data sets, these are generally smaller
than the differences between the reanalyses and the Loon data. High frequency variability at periods close to and below one
day is more noticeable in the Loon observations than in any of the reanalyses which suggests that these small-scale variations
might be important in explaining any differences. The differences likely represent the impact of small–scale waves, with a
number of studies identifying that inertia-gravity waves may be important.

Statistics of the reanalyses minus Loon-derived wind differences, over a wide range of southern latitudes, show that the
Loon-derived wind fields match well with the reanalyses. Histograms and key statistics of the wind differences are shown in
Fig. 3 and Table 2. The wind differences shown in Fig. 3 all exhibit Gaussian distributions with biases less than 0.37 m/s and
standard deviations less than 3.4 m/s. These values are larger than those derived by Boccara et al. (2008) who found zonal
and meridional standard deviations of 2.43 and 2.38 m/s for the differences between ECMWF operational analyses and the
Vorcore-derived winds. However, the larger standard deviations derived in our study are consistent with the observed latitudinal
trend for the standard deviation as discussed below. Table 2 also shows that the mean zonal wind difference between the Loon-
derived winds and the reanalyses is larger for ERA–Interim and CFSv2 than for MERRA and MERRA-2. It is also clear that
inter-reanalyses differences in the standard deviations of the zonal and meridional wind differences are small.

The latitudinal structure in the differences between the Loon and reanalyses winds, shown in Fig. 4, shows a tendency for
the standard deviation in the wind differences to increase closer to the equator. Although there is no obvious trend in the zonal
wind biases, ERA-Interim has a consistent positive bias over all latitude ranges as opposed to the biases in the other reanalyses
which switch sign. The large ERA-Interim zonal bias statistic listed in Table 2 is therefore not an indicator that ERA–Interim
is worse in this respect than the other reanalyses, but rather that it exhibits a consistent bias across latitude whereas the other
reanalyses have biases of similar magnitudes which cancel when averaged over latitude. Across all reanalyses, there appears to
be a trend in the meridional biases with net over-estimation polewards of $\sim 40°$S and under-estimation equatorward of $\sim 40°$S
.

While the region closest to the equator has larger biases and standard deviations, these biases are significantly smaller than
those derived by Podglajen et al. (2014). This may be related to seasonal differences, where most of the Loon flight data were
collected through the Southern Hemisphere winter (June to September), while the measurements analyzed by Podglajen et al.
(2014) were collected in February. However, given the lack of strong seasonal variations in the tropics, this inference is ques-
tionable. Another possibility is that inter-annual variability in the mean winds could play a significant role; the phase of the
Quasi–Biennial Oscillation could be important. The work in Podglajen et al. (2014) also highlighted large wind biases in spe-
cific regions (i.e. the Indian Ocean and the Eastern Pacific) where in situ observations are scarce. Therefore, given the limited





quantity of observations near the equator in both studies, we cannot exclude the effects of sampling bias between the two data sets.

The wind difference statistics indicate that of the four reanalyses analyzed, ERA-Interim and MERRA-2 perform the best with MERRA-2 showing a measureable improvement over MERRA.

## 3.2 Trajectories

The trajectory model described above was used to initialize a simulated trajectory every six hours along the observed Loon balloon trajectory. The resultant separation statistics between the observed and simulated trajectories are shown in Fig. 5 and Table 3. The mean and median values of the trajectory separations as a function of time are shown in panel (a) of Fig. 5 for the four different reanalyses. A more detailed representation of the separation of the trajectories calculated from the MERRA-2 wind fields from the observed trajectories is shown in panel (b) of Fig. 5, including confidence intervals and inter-quartile ranges.

If a trajectory's corresponding balloon underwent rapid altitude changes over the course of the simulated trajectory, only the separation data up to that altitude change are included, resulting in a decreasing number of available trajectories as time progresses (Fig. 5 (c)). The results plotted in panel (a) of Fig. 5 show that after the first day, both the mean and median separations increase roughly linearly with time. For MERRA-2, the median separation grows at a rate of roughly 48 km a day. However, the growth of individual trajectory separations is far more chaotic. The departures between the mean and median values of the separation at a particular time along the trajectory suggest there are significant contributions due to extreme outliers, with the mean approaching the upper quartile of separations (Fig. 5 (b)). This also suggests that the median is likely a better indicator of expected trajectory separation. Histograms of the 5-day separations between the reanalyses-based simulations and the Loon trajectories are displayed in Fig. 6. After 5 days, the separations resulting from the MERRA-2-derived trajectories show a smaller number of large outliers and also a slightly higher proportion of simulations at lower separations than the other three reanalyses (Fig. 6). The histograms display a roughly log-normal distribution. A log-normal process is the statistical realization of the multiplicative product of many independent positive random variables, this form is therefore suggestive of the fact that a combination of multiple factors impacts the separations observed. Comparison between the MERRA and MERRA-2 distributions also shows that the MERRA-2-based trajectories follow more closely the actual Loon trajectories.

The separation statistics shown in Fig. 5 compare well with the analyses detailed in Hertzog et al. (2004) and Boccara et al. (2008) although, surprisingly, the ECMWF analyses used in Hertzog et al. (2004) have somewhat smaller separations at 5 days than those in this study. This may result from the higher quality of reanalyses in the Northern Hemisphere relative to the Southern Hemisphere identified in some previous studies. That said, given the improvement in the quantity of data being assimilated by the more recent reanalyses, and underlying model improvements, this is still a little puzzling.

If trajectories after forced balloon altitude manoeuvres are not excluded from the analyses, we find that the comparisons of the observed and modelled trajectories decrease significantly in quality. The median MERRA-2 separation after 5 days increases from 240 km to 574 km, increasing at a rate of roughly 88 km per day. This increase could be expected as trajectories



that were initially separated due to small biases in reanalyses, but still follow along the same general flow, might suddenly find themselves in different flow regions when the pressure level is adjusted, leading to higher trajectory separations. However, this apparent degradation in trajectory quality could also be an indicator of selection bias. The Loon team uses numerical weather prediction (NWP) model output to forecast balloon trajectories, and any balloon motion not predicted by the NWP might

require adjustment using forced altitude changes. This would then result in our analysis excluding the effects of the long-term behaviour of these inaccurate trajectories. Similarly, if the reanalyses have difficulty modelling these trajectories, this would lead to an automatic selection bias with the long-term separation statistics including more 'good' trajectories. The short-term separation statistics are likely to be more reliable and less prone to this sampling bias.

To examine the separations in an alternative manner, we can also inspect the relative separations. There are two variants of
this approach. We can examine the separation at some time divided by the total distance travelled by the balloon over 8 days, or alternatively, the separation after $h$ hours divided by the distance travelled by the balloon during those $h$ hours. One motivation for the former method is that if trajectories that travel further have concomitant greater separations, this might diminish the effect of these outliers. The resulting relative separations are shown in Fig. 7. A notable feature in the first relative separation method is that the MERRA-2 and ERA–Interim mean relative separations are much more distinct, and that the mean relative
separations of the reanalyses are much closer to the median, lying well within the inter-quartile ranges. The second method also shows some interesting features, with median relative separations remaining roughly constant after the first day, for example the MERRA-2 shows a consistent median relative separation of $\sim 10\%$.

Comparison of the results from Fig. 5 and 7 (a) suggests that the trajectories with the highest separations tend to correspond to the flights with the longest distances travelled, also revealed when performing a more in depth examination of individual
events. In particular, there is a low correlation ($r = 0.34$) between total distance travelled and the resulting separation, but the mean separations for the upper-half of distance-traveled-balloons is nearly double of the lower half suggesting that this factor might dominate the observed variations. This would suggest that while the differences between the reanalyses and Loon winds are important in defining the separation, the mean state of the wind also plays an important role, as one would expect. In addition, the difference in separation statistics between the ERA-Interim and MERRA-2 could then be identified to be related
to the larger bias in the zonal mean in the ERA-Interim than the MERRA-2 dataset.

There is little latitudinal variation in trajectory accuracy, but we do find that for all reanalyses the mean trajectory separations are slightly lower between $15°S$ and $35°S$ than between $35°S$ and $55°S$. This is slightly counter-intuitive because the standard deviation of wind errors display the opposite trend. This is likely explained by the fact that the growth of the separation depends on the type of flow, for example, over eight days the balloon trajectories tend to travel a greater total distance as the latitude
increases, which might explain the observed trend in trajectory accuracy. For the relative separation, separation divided by total distance travelled, shown in Fig. 7 the opposite trend is observed with greater separations equator-ward.

Notably, we find that the MERRA-2 trajectories are significantly improved with respect to the old MERRA version 1 trajectories, resulting in trajectories with similar mean separation statistics to those derived from ERA-Interim. While the mean separations are nearly indistinguishable, the MERRA-2 median separation is noticeably lower than that of ERA–Interim sug-
gesting that the MERRA-2 separation distribution is more skewed than that of the ERA–Interim.





### 3.3 Temperature

There are several difficulties associated with the Loon temperature data. As previously stated, the data is a measurement of the lift gas temperature and not of the ambient air, resulting in a strong solar zenith angle (SZA) dependent bias. Additionally, although we are not aware of the specific instruments used, it seems that the thermometer used has a high uncertainty and is

5 intended as a diagnostic instrument rather than for scientific data collection. An example of balloon–reanalysis temperature differences is show in Fig. 8. The SZA bias can be corrected through the use of a correction function, as is commonly done to correct balloon based temperature measurements (Hertzog et al., 2004, 2006; Knudsen et al., 2006), but it should be noted that the impact of solar heating on the lift gas temperature is much more significant than the usual solar bias, up to $+30$ K as opposed to the typical $\sim 1.5$ K. The SZA dependent bias can be modelled as:

$$
\quad T_{bias} = \alpha \quad + \begin{cases} \beta(1 - e^{(\theta-95)/\lambda_0}) + \gamma e^{-(\theta-90)^2/\lambda_1} & \theta \leq 90 \\ \beta(1 - e^{(\theta-95)/\lambda_0}) + \gamma e^{-(\theta-90)^2/\lambda_2} & 90 < \theta \leq 95 \\ \gamma e^{-(\theta-90)^2/\lambda_2} & 95 < \theta \leq 150 \\ \gamma e^{-(\theta-90)^2/\lambda_2} + \delta \cdot (\theta - 150) & 150 < \theta \end{cases} \quad (1)
$$

where $\alpha$, $\beta$, $\gamma$, $\delta$, $\lambda_0$, $\lambda_1$ and $\lambda_2$ are fit coefficients determined from a linear least squares regression. After removing some flights with anomalous observations (unreasonably large bias, questionable GPS or pressure data), we use temperature data from every second flight to fit the correction function, and then apply this correction to the remaining flights. The fitted parameters are provided in Table 4, and Figure 9 shows the CFSv2 temperature differences with and without the correction

applied. Application of the correction functions reduces the mean Loon-reanalyses temperature differences to a few degrees, significantly improving the utility of the Loon temperature measurements, however the standard deviation and the shorter term, day by day biases are still much greater than observed in other studies.

Ignoring the SZA bias by focusing only on the nighttime measurements, we still find a standard deviations of $\sim 6$ K while other balloon studies typically have biases and standard deviations less than $2$ K. Additionally the nighttime measurements

show interesting behaviour with common consistent night-long biases of up to $\pm 10$ K. Considering the upper bound on the thermometer uncertainty provided by the Loon team, the significant SZA bias which is much greater than those usually dealt with using correction functions, and the unusually inaccurate night-time temperatures, leads us to conclude that currently the quality of the Loon temperature data means it is of little value in assessing the quality of the reanalyses. Particularly, the variations in the differences between the reanalyses and the corrected temperatures is dominated by the uncertainty in the

temperature observations, as the reanalyses show only a $\sim 0.2K$ variation in the biases and standard deviations.

### 4 Discussion and Conclusions

Loon long-duration balloon GPS trajectory information has been used to examine the quality of the horizontal winds in re-analyses along with the concomitant trajectory errors. The fundamental goal of this study is to test the potential for the Loon balloons to be used in the evaluation of reanalyses fields in the stratosphere. This dataset is potentially of high value because





with the exception of the EOLE experiment detailed in Hertzog et al. (2006) the number of measurements available in previous studies has been far fewer than the current dataset. It should also be noted that the EOLE experiment took place in 1971-1972 and therefore occurred previous to the satellite era and thus potentially does not offer a good test of the quality of the reanalyses given the very limited amount of data that was assimilated in the Southern hemisphere before the satellite era. Our results are

generally in agreement with the limited number of previous studies. In particular, we find differences between reanalyses winds and the winds derived from the Loon trajectories that are comparable with those in Knudsen et al. (2006) and Boccara et al. (2008), these differences are also smaller than those identified Podglajen et al. (2014) but slightly larger than those identified in Hertzog et al. (2004). In this study, latitude dependent wind biases of less than $0.5\,\mathrm{m/s}$ and standard deviations of roughly $3\,\mathrm{m/s}$ are observed. In common with Hertzog et al. (2006) and Podglajen et al. (2014) we also find that the standard deviation

of these differences increase toward the equator. We also note that these Southern hemisphere measurements have larger differences with the reanalyses than identified in the Northern hemisphere study detailed in Hertzog et al. (2004). Unfortunately, we also find that currently the Loon temperature measurements are not suitable for comparison with reanalyses even after a correction scheme similar to the one developed in Hertzog et al. (2004) is applied to the data. When considering the biases and standard deviations linked to the four reanalyses used in this study (ERA-Interim, MERRA, MERRA-2 and CFSv2) we

find that ERA-Interim and MERRA-2 have slightly smaller standard deviations than the other two products. The improvement between the MERRA and MERRA-2 reanalyses being a notable achievement.

When the trajectories derived from the reanalyses winds are compared to the balloon trajectories, we again find broad comparability with previous studies. For example, the resulting 5 day mean (median) trajectory separations are found to vary from 620 (320) to 760 (480) $\mathrm{km}$ while work detailed in Boccara et al. (2008) found mean spherical distances between 400

and 1000 km after 5 days. We also note that the present results are somewhat better than those identified in Knudsen et al. (2006) (1300 km after 5 days) which might be a little surprising given that inspection of Figure 2 in that paper suggests the standard deviations in the winds used in the trajectory model are comparable. However, a larger bias in the zonal wind (0.7m/s) was identified in Knudsen et al. (2006) than in the current study. We also note that the detailed methodology used in the current study and Knudsen et al. (2006) are very similar and we therefore suggest that this difference may be associated

with latitudinal differences in the quality of the reanalyses. It is also notable that MERRA version 2 performs the best out of all the examined reanalyses, showing significant improvements over version 1. The relative separation analysis detailed in Fig. 7 is also suggestive that the mean state and therefore the distance travelled by the balloon plays a role in these separation statistics. This fact likely explains the latitudinal structure of the trajectory statistics, with marginally lower mean separations between $15°S$ and $35°S$ than between $35°S$ and $55°S$ in all four reanalyses despite standard deviations in the wind differences

increasing toward the equator.

As it stands, balloons launched as part of the X Project Loon network provide a useful independent test of atmospheric reanalysis winds. More balloons will continue to be launched which, if they are not assimilated into reanalyses, will allow significantly greater coverage for reanalysis comparisons, and perhaps enable an investigation into the seasonal variability of reanalysis accuracy. Further opportunities for understanding the mixing in the stratosphere using the currently available Loon

data are also being currently explored.





*Acknowledgements.* The work discussed would have been impossible without support from the New Zealand Antarctic Research Institute. We also thank the X Project Loon team for the generous supply of the Loon data.



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



**Table 1.** Resolution of the model outputs used in this study. All model products provided in six hour intervals.

|            | Latitude | Longitude | Pressure levels |
|------------|----------|-----------|-----------------|
| ERA–Interim | 3/4°    | 3/4°      | 37              |
| MERRA      | 1/2°     | 2/3°      | 42              |
| MERRA-2    | 1/2°     | 5/8°      | 42              |
| CSFv2      | 1/2°     | 1/2°      | 37              |

Pawson, S., Pegion, P., Redder, C. R., Reichle, R., Robertson, F. R., Ruddick, A. G., Sienkiewicz, M., and Woollen, J.: MERRA: NASA's Modern-Era Retrospective Analysis for Research and Applications, J CLIMATE, 24, 3624–3648, 2011.

Saha, S., Moorthi, S., Wu, X., Wang, J., Nadiga, S., Tripp, P., Behringer, D., Hou, Y.-T., ya Chuang, H., Iredell, M., Ek, M., Meng, J., Yang, R., Mendez, M. P., van den Dool, H., Zhang, Q., Wang, W., Chen, M., and Becker, E.: NCEP Climate Forecast System Version 2 (CFSv2) 6-hourly Products, http://dx.doi.org/10.5065/D61C1TXF, accessed: 2015-12-9, 2011.

Smith, M. L. and McDonald, A. J.: A quantitative measure of polar vortex strength using the function M, J GEOPHYS RES-ATMOS, 119, 5966–5985, doi:10.1002/2013jd020572, <GotoISI>://WOS:000337766600014, 2014.

Stohl, A., Cooper, O. R., and James, P.: A cautionary note on the use of meteorological analysis fields for quantifying atmospheric mixing, J ATMOS SCI, 61, 1446–1453, 2004.

Sutton, R. T., Maclean, H., Swinbank, R., Oneill, A., and Taylor, F. W.: High-resolution stratospheric tracer fields estimated from satellite-observations using lagrangian trajectory calculations, J ATMOS SCI, 51, 2995–3005, doi:10.1175/1520-0469(1994)051<2995:hrstfe>2.0.co;2, 1994.

Tuck, A. F., Hovde, S. J., Gao, R. S., and Richard, E. C.: Law of mass action in the Arctic lower stratospheric polar vortex January-March 2000: ClO scaling and the calculation of ozone loss rates in a turbulent fractal medium, J GEOPHYS RES-ATMOS, 108, –, 2003.

Vondergathen, P., Rex, M., Harris, N. R. P., Lucic, D., Knudsen, B. M., Braathen, G. O., Debacker, H., Fabian, R., Fast, H., Gil, M., Kyro, E., Mikkelsen, I. S., Rummukainen, M., Stahelin, J., and Varotsos, C.: Observational evidence for chemical ozone depletion over the arctic in winter 1991-92, NATURE, 375, 131–134, 1995.



**(a)**

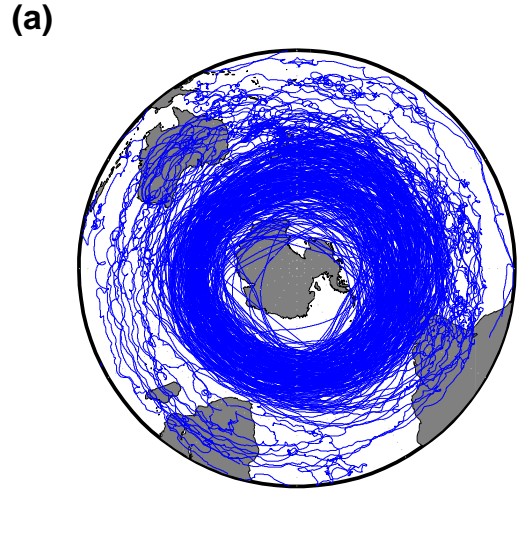

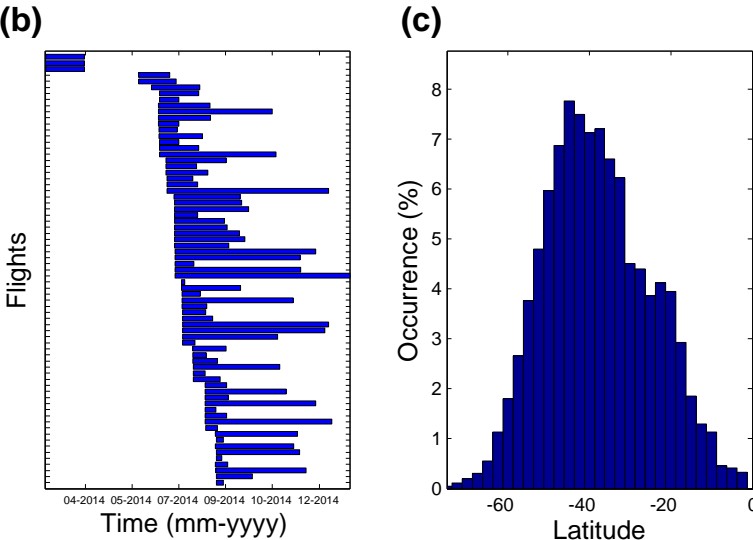

**Figure 1.** General Loon flight information including (a) set of all balloon trajectories viewed from south pole, (b) time-line showing individual balloon launch times and flight durations, and (c) histogram of observation distribution as a function of latitude.





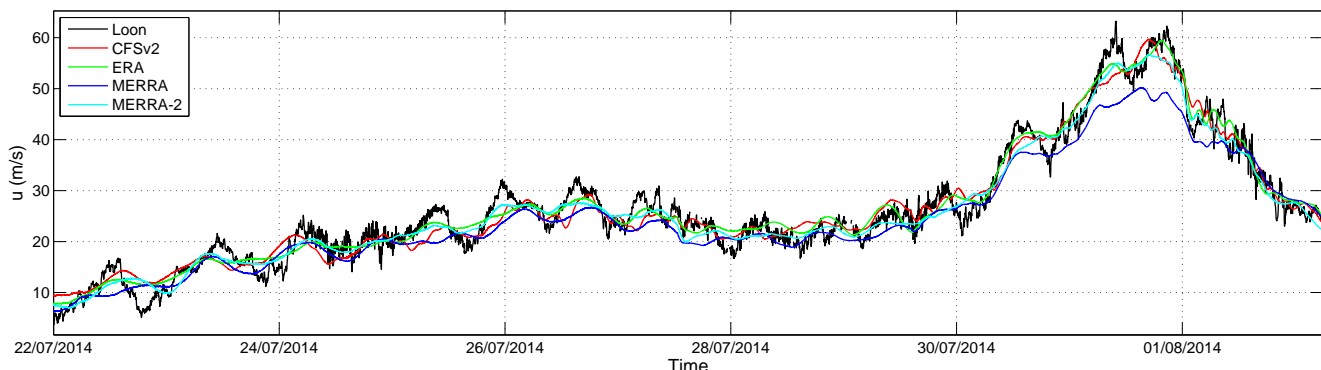

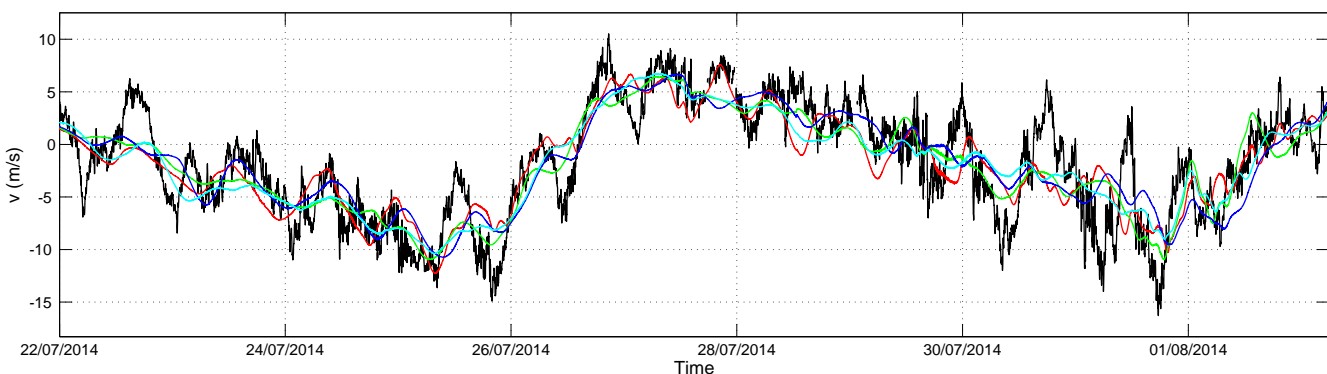

**Figure 2.** Wind speeds measured from Loon flight # 263 along with interpolated reanalysis winds. This shows the typical behaviour for comparisons of balloon and reanalysis wind speeds, including the tendency for the balloon winds to oscillate about the reanalysis winds.

**Table 2.** Statistics of the wind differences between the reanalyses and the Loon balloons. Corresponding histograms are plotted in Fig. 3. Units are m/s

|  | ERA–Interim | CFSv2 | MERRA | MERRA-2 |
|---|---|---|---|---|
| $u_{\mathrm{diff}}$ Mean | 0.3662 | 0.2204 | -0.0090 | 0.0992 |
| $v_{\mathrm{diff}}$ Mean | 0.0502 | 0.0462 | 0.0747 | 0.0671 |
| $u_{\mathrm{diff}}$ SD | 2.8609 | 3.1378 | 3.1254 | 2.9090 |
| $v_{\mathrm{diff}}$ SD | 3.1481 | 3.3522 | 3.3199 | 3.1817 |
| $u_{\mathrm{diff}}$ Skewness | 0.1173 | 0.0741 | -0.0230 | 0.0969 |
| $v_{\mathrm{diff}}$ Skewness | 0.0281 | 0.0224 | 0.0268 | 0.0149 |



**Figure 3.** Zonal and meridional wind difference histogram outlines. Histograms are binned by steps of $0.25\ \mathrm{m/s}$. Corresponding statistics are shown in Table. 2.

**Table 3.** Statistics of the trajectory separations after five days in $\mathrm{km}$. Corresponding Separations over time plots are provided in Fig. 5. The errors on the means are the $90\%$ confidence intervals.

|  | ERA–Interim | CFSv2 | MERRA | MERRA-2 |
|---|---|---|---|---|
| Mean | $638 \pm 29$ | $661 \pm 30$ | $764 \pm 33$ | $625 \pm 34$ |
| Median | 381 | 415 | 486 | 327 |





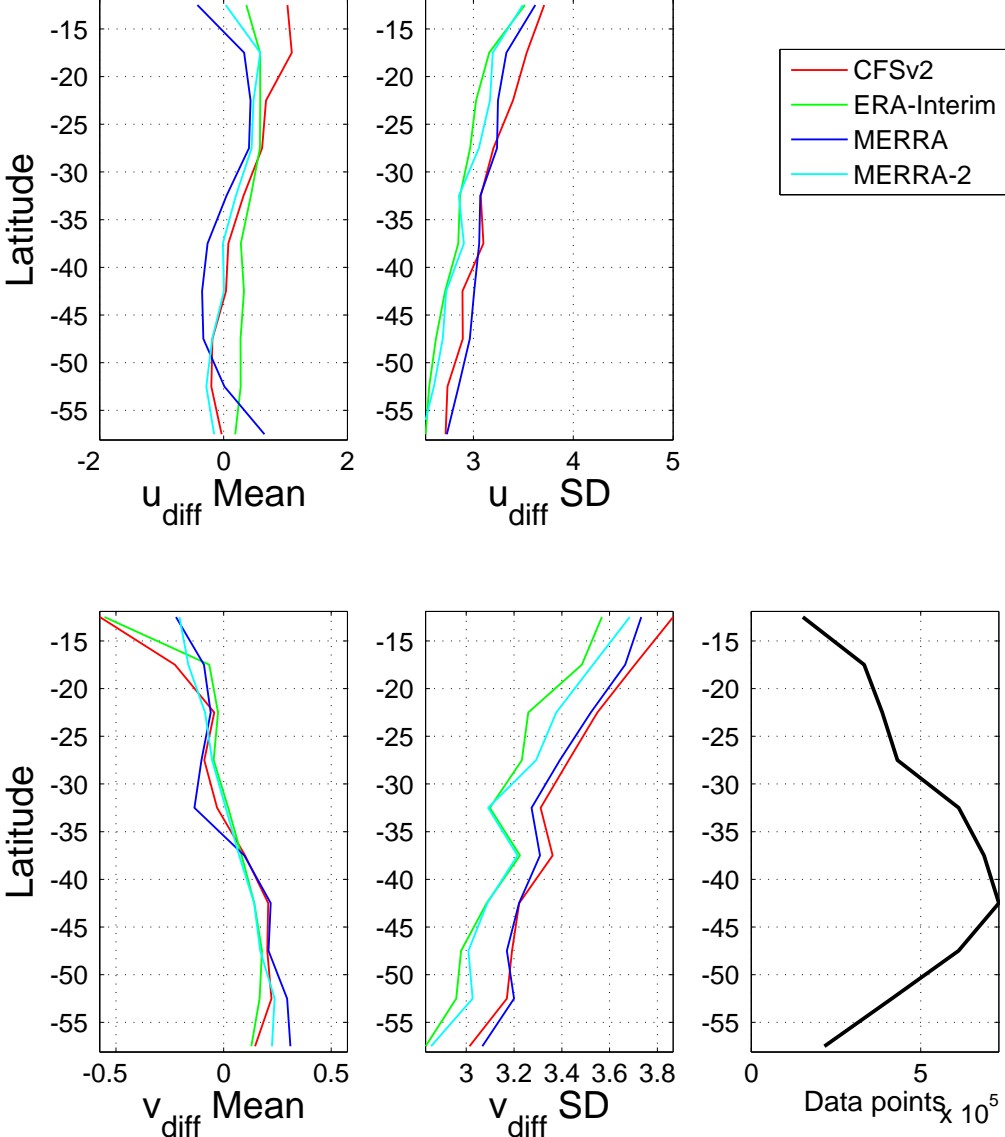

**Figure 4.** Zonal and meridional wind differences binned by latitude, in $1°$ steps. There is a clear tendency for wind difference standard deviations to be larger near the equator. There also seems to be a trend in the meridional wind differences, with net over (under) estimation poleward (equatorward) of $40°$ S.





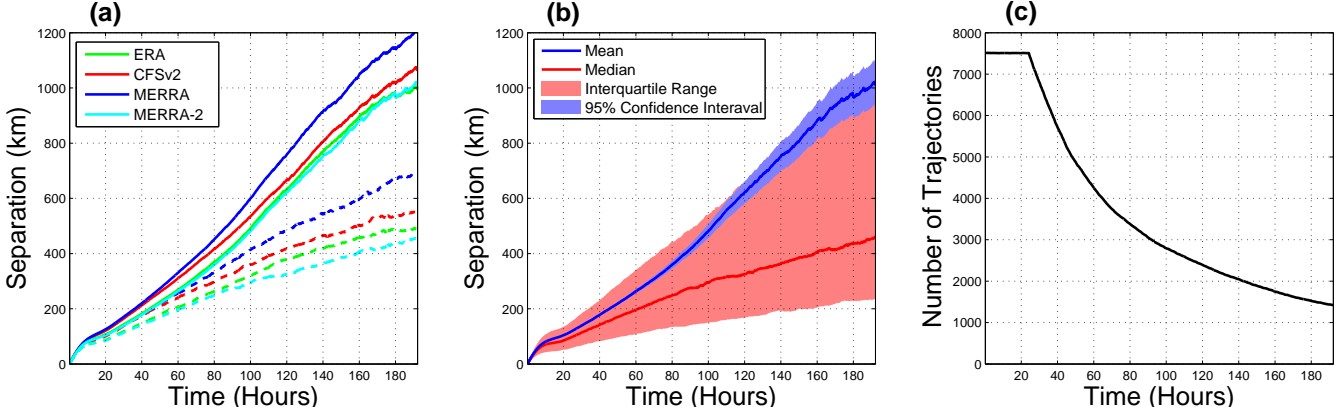

**Figure 5.** Trajectory separations as a function of time. (a) shows a comparison of the trajectory quality of each of the reanalyses with solid lines representing the mean and dashed lines the median separations. (b) provides a more detailed plot of the MERRA-2 trajectories, including confidence intervals and inter-quartile ranges. More detailed plots for the other reanalyses show very similar characteristics to those observed in (b). (c) provides information on the number of trajectories included at each hour mark, decreasing due trajectories running over altitude changes.

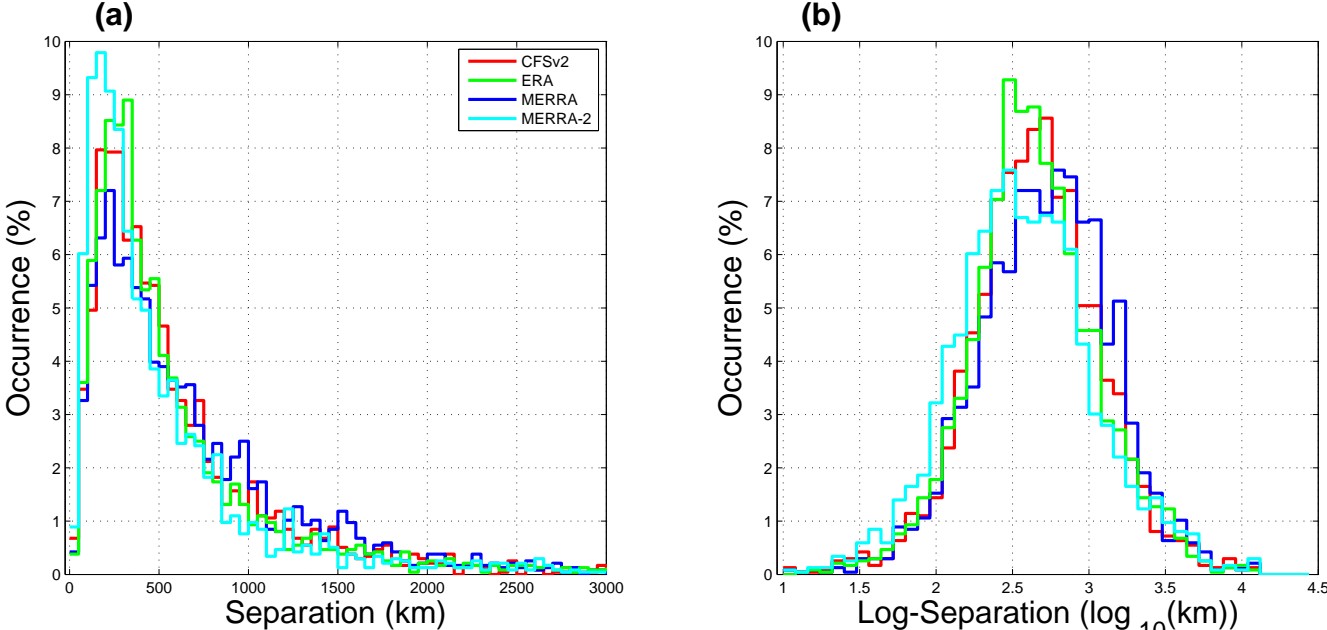

**Figure 6.** Histogram of the trajectory separation distribution after 5 days. (b) is the same as (a), but using logarithmic separation to highlight the log-normal distribution, with a long tail of extreme outliers which is not visible in (a).



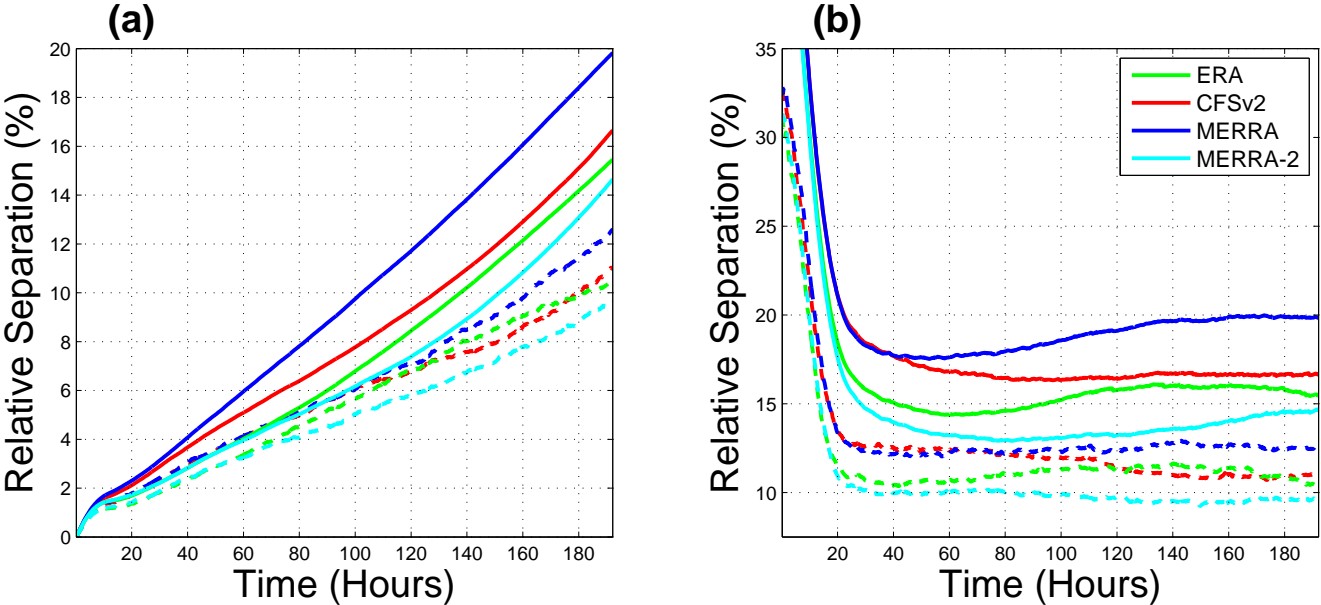

**Figure 7.** Relative trajectory separations as a function of time. (a) is similar to (a) in Fig. 5 except here, prior to deriving the statistics, the separation of each reanalysis trajectory is normalized by the total distance traveled by the balloon during those eight days. (b) is similar, except here the separations are divided by the distance the balloon has traveled at each our mark, not the total.

**Table 4.** Best fit correction function parameters as determined by applying the correction to every second flight.

| $\alpha$ | $\beta$ | $\gamma$ | $\delta$ | $\lambda_0$ | $\lambda_1$ | $\lambda_2$ |
|---|---|---|---|---|---|---|
| $-0.4116$ | $28.25$ | $5.039$ | $0.2345$ | $21.39$ | $113.5$ | $13.76$ |



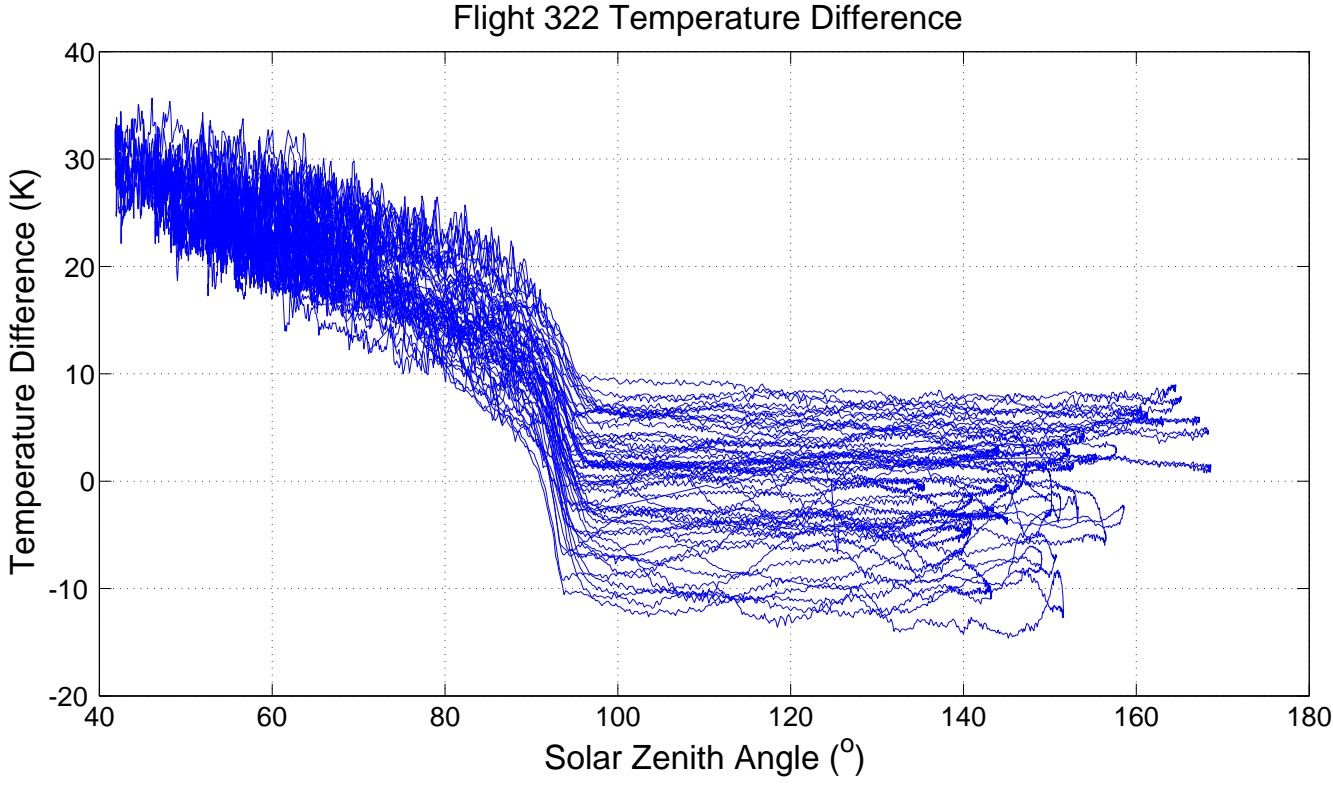

**Figure 8.** Differences between Loon lift gas and interpolated MERRA-2 temperatures for flight 322. The SZA dependent bias is clearly visible





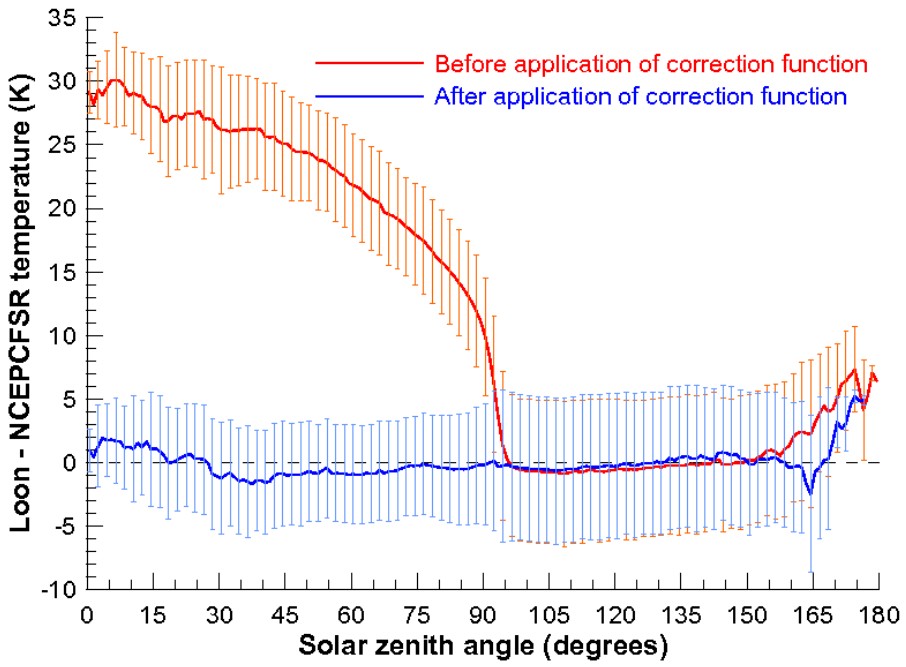

**Figure 9.** Differences between Loon lift gas temperatures obtained from selected odd-numbered flights (red traces) and temporally and spatially coincident NCEPNCAR (top panel (a)) and NCEPCFSR (bottom panel (b)) reanalysis temperatures. Mean differences in each 1° SZA bin are shown with a solid line together with the first standard deviation of the differences as uncertainty bars. Differences after the application of the correction functions are shown in blue.