# Peer review of "A comparison of Loon balloon observations and stratospheric reanalysis products"

_Atmospheric Chemistry and Physics, 2016_

## Referee Comment (RC1) · Anonymous Referee #1 · 16 Jun 2016

In this paper, measurements gathered during stratospheric long-duration balloon flights performed in the frame of Google Loon project are compared to reanalysis products. These measurements were not assimilated by Numerical Weather Prediction systems and thus provide an independent dataset that can be used to assess reanalysis accuracies. The 70 Loon Balloon used in this study flew in the Southern Hemisphere lower stratosphere. The study focuses on wind and trajectory comparisons since primary observations provided by Loon balloons are balloon positions, from which horizontal wind components are derived. Since reanalysis products are widely used to e.g. study transport processes in the lower stratosphere, this study is particularly relevant to have independent information on their accuracies, which is otherwise difficult to get with more classical datasets that are generally assimilated.

I have found that the material and figures of this article are generally well presented,

and in my mind, the article adresseses topics that are of much interest to the ACP readership. Furthermore, one can hope that Google Loon will continue flying long-duration balloons in the future, and such study is particularly useful to demonstrate the potential of observations obtained with such flights. Yet, I have the impression that the article could be significantly improved if, in several instances, further information were provided. I also think that there is a flaw in how lift-gas temperature measurements are treated in this study. I would therefore encourage the authors to carefully address my remarks below, and would recommend publication afterwards.

**Main issues**

1. Balloon dataset: I would appreciate if you could provide (perhaps in Figure 1) an histogram of balloon pressures and altitudes. It is important to know whether the balloon measurements are representative of a specific thin layer of the atmosphere or do indeed provide homogeneous information on the 30-70 hPa layer as stated in p4, l15.

2. Ballon vertical excursion: in p4 l21, it is stated that "whenever a pressure change greater than $5$ hPa occurs within one hour, the balloon is considered to be undergoing an altitude control manoeuvre". Could you provide an illustration of either pressure or balloon altitude timeseries that shows such manoeuvre, and clearly displays which part of the dataset is discarded?

3. Sensor precisions: observations performed on Google balloons were likely not primary intended to provide scientific-class measurements, and stated sensor precisions (p4 l28) are rather large compared with current state-of-the-art meteorological measurements. This is not an issue in itself provided that the impacts of the fairly large measurement uncertainties are precisely assessed. This aspect needs to be improved in the current manuscript:

- First, if one assumes that the uncertainty on the GPS horizontal position is $10$ m (as mentioned in p4, l28), and furthermore that the uncertainties on the two positions separated by $\Delta t$ that serve to compute the winds are independent (which is not explicitly stated), then the uncertainty on the derived wind should be $10\sqrt{2}/\Delta t = 0.23\,m\,s^{-1}$, with $\Delta t = 1$ min (p26, l4). It is only when $\sqrt{2}$ is replaced by $2$ than one comes to the $0.33\,m\,s^{-1}$ value reported in the paper, which I do not understand.

- The pressure measurement is used as the vertical coordinate in the interpolation of the reanalysis product onto the balloon position. As stated in the paper (p4., $l28 - 31$), a $1.5$ hPa uncertainy in these measurements is "rather large" and "could potentially lead to uncertainties when vertically interpolating the reanalysis data sets to the balloon locations". While likely true, this sentence stays very qualitative. It would be much helpful if a typical vertical wind shear could be assumed so as to infer a resulting uncertainty on the interpolated wind.

When all these measurement/interpolation uncertainties are properly taken into account, one can better know which part of the differences between the balloon observation and analysis is due to the observations or to defficiencies in the analysis (Section 3.1, and Figures 3 and 4).

4. Could you also provide confidence intervals in Figure 4, and state which values are significant in Table 2? And please provide only significant digits in this table.

5. One way to identify the uncertainty on the wind measurements is to compute the spectrum of wind disturbances and look where the spectrum becomes flat at high frequencies. The raw timeseries could then even be filtered to eliminate the high-frequency noise, and comparisons with the reanalyses could be made with these filtered timeseries, which would more accurately estimate the analysis defficiciencies. I therefore think that providing the wind spectrum would be a very

valuable addition to the article.

6. As mentioned on p5, l3 − 5, the temperature measurements provided by Loon balloons are those of the "lift gas", and not of the ambient air. In Section 3.3, the authors use an empirical method to correct the lift-gas temperatures from observed diurnal variations that are implicitly assumed to be spurious, and claim that "this [method] is commonly used to correct balloon based temperature measurements". While it is true that such method has been previously used (articles cited in the paper), it was solely used to correct air temperature observations in the lower stratosphere. Its use to correct lift gas temperature measurements, as done here, is more questionnable: one assumption of this method is indeed that the undelying 'true' temperature is not exhibiting diurnal variations (or that the diurnal cycle is less than the sensor uncertainty). I doubt that this is the case for the lift gas temperature: the balloon envelop certainly absorbs to some extent the sun radiations, which would unavoidably lead to an increase of the gas temperature during day. It is certainly true that the temperature sensor itself absorbs these radiations, and thus overestimates the gas temperature diurnal cycle. But correcting the measurements to fully eliminate the diurnal cycle is likely excessive. I would thus recommend to discard using the temperature correction, but I would keep Figure 8, and slightly rephrase the sentence on page 9 l 20: it is not only the "quality of the Loon temperature data" which is an issue, it is also the fact that they only measure the gas temperature, which can be quite different to that of the air.

**Minor points**

- p2 l35 to p4 l9 is a long report on previous studies that used similar methodology than the one used in this study. I do not discuss the interest of mentioning these various studies to motivate the present work. I nevertheless think that the discussion could be somewhat synthetized and maybe re-organized by e.g. Earth regions (Northern Hemisphere high latitudes, tropics, Southern hemisphere mid/high latitudes), so as to ease the reader to get a clear picture of these previous results.

- end of introduction: could you provide the plan of your study here?

- p6, l10: could you be more specific on the studies that attribute differences to inertia-gravity waves? Could you furthermore state at which latitude the time-series displayed on Figure 2 were obtained? The frequency of inertia-gravity wave depends on latitude, and it may be worth testing that the apparent period of the short timescale disturbances in the wind timeseries indeed corresponds to the inertial period.

- Sentences on p6 l18 and p7 l3 do not seem to be consistent: does ERA-Interim performs better than MERRA?

---

## Referee Comment (RC2) · Anonymous Referee #2 · 28 Jun 2016

Review of "A comparison of Loon balloon observations and stratospheric reanalysis products" by Friedrich et al.

<General comments>

The authors compared zonal and meridional winds of ERA-Interim, MERRA, MERRA–2 and NCEP CFSv2 reanalyses with those derived from X Project Loon. The goal of this study is to test the potential for the Loon to be used in the evaluation of reanalyses. Because the Loon data is not assimilated in each reanalysis, the Loon should provide a useful independent test of atmospheric reanalysis winds.

The purpose of this study is well written. This paper includes very new results contributing to reanalysis communities. X Project Loon might have large potential to improve reanalyses in the future. I believe this paper is suitable for the publication in ACP. I

have a few comments written below.

\<Specific comments\>

(1) Abstract: "All reanalysis data sets accurately describe the winds, with biases in zonal winds of less than 0.37m/s and meridional biases of less than 0.08m/s. The standard deviation on the differences between Loon and reanalyses zonal winds is latitude dependent, ranging between 2.5 and 3.5 m/s increasing equatorward"

P3L2–3: "Both analyses accurately represented the winds with biases of less than 0.3 m/s and standard deviations ranging from 2.3 to 2.7 m/s", and also P3L8, P3L14, P3L18, P4L3–5 etc.

The bias and standard deviation depend on time scales analyzed (i.e., hourly, daily, monthly). The authors should indicated output intervals analyzed in each case. In addition, as the authors sometimes speculate differences due to gravity waves, differences between reanalyses and observations also depend much on their vertical resolutions. More careful explanations are needed.

(2) Kawatani et al. (2016) may be useful for this paper, which shows the standard deviations calculated by monthly mean data among several reanalyses show large values in the equatorial stratosphere, and their geographical distributions in the lower stratosphere are closely related to the density of in-situ radiosonde observations.

Kawatani, Y., Hamilton, K., Miyazaki, K., Fujiwara, M., and Anstey, J. A.: Representation of the tropical stratospheric zonal wind in global atmospheric reanalyses, Atmos. Chem. Phys., 16, 6681-6699, doi:10.5194/acp-16-6681-2016, 2016.

(3) P4L21: The authors mention the balloons is considered to be undergoing an altitude control whenever a pressure change greater than 5 hPa occurs within one hour.

Providing height information of Loon flights in Fig. 1 should be useful for reads to imagine.

(4) P4L29–31: "The upper bound on the pressure sensor uncertainty is rather large and could potentially lead to uncertainties when vertically interpolating the reanalyses data sets to the balloon locations"

I guess uncertainties depend much on the wind profiles as the vertical resolutions of reanalyses are coarse. Could you specify uncertainties more quantitatively?

(5) Table 1: As the top boundary of each reanalysis is different, showing layer numbers between 30 and 70hPa (i.e., the varying pressure levels of the balloon flights) should be useful.

(6) P6L31–33: How the phases of the Quasi-Biennial Oscillation play the role of difference between this study and Podglajen et al. (2014)? The authors could provide their speculation in more details.

---

## Author Comment (AC1) · 16 Sep 2016

**We would like to thank the reviewers for their insightful questions and suggestions for improvement that they have provided. In the remainder of this document bold text identifies responses to reviewers comments.**

**Author team.**

**Reviewer 1 Comments and Responses** In this paper, measurements gathered dur-

ing stratospheric long-duration balloon flights performed in the frame of Google Loon project are compared to reanalysis products. These measurements were not assimilated by Numerical Weather Prediction systems and thus provide an independent dataset that can be used to assess reanalysis accuracies. The 70 Loon Balloon used in this study flew in the Southern Hemisphere lower stratosphere. The study focuses on wind and trajectory comparisons since primary observations provided by Loon balloons are balloon positions, from which horizontal wind components are derived. Since reanalysis products are widely used to e.g. study transport processes in the lower stratosphere, this study is particularly relevant to have independent information on their accuracies, which is otherwise difficult to get with more classical datasets that are generally assimilated. I have found that the material and figures of this article are generally well presented, and in my mind, the article adresseses topics that are of much interest to the ACP readership. Furthermore, one can hope that Google Loon will continue flying longduration balloons in the future, and such study is particularly useful to demonstrate the potential of observations obtained with such flights. Yet, I have the impression that the article could be significantly improved if, in several instances, further information were provided. I also think that there is a flaw in how lift-gas temperature measurements are treated in this study. I would therefore encourage the authors to carefully address my remarks below, and would recommend publication afterwards.

Main issues 1. Balloon dataset: I would appreciate if you could provide (perhaps in Figure 1) an histogram of balloon pressures and altitudes. It is important to know whether the balloon measurements are representative of a specific thin layer of the atmosphere or do indeed provide homogeneous information on the 30-70 hPa layer as stated in p4, l15.

**A histogram of the distribution of pressure observed over the entire flight period has been added as panel (d) of Figure 1 to address this point. See the updated**
* * *
**Figure (Figure 1 at the end of this document) and the updated caption below.**

**Figure 1: General Loon flight information including (a) set of all balloon trajectories viewed from south pole, (b) time-line showing individual balloon launch times and flight durations, (c) histogram of observation distribution as a function of latitude, and (d) histogram of observation distribution as a function of pressure.**

2. Ballon vertical excursion: in p4 l21, it is stated that "whenever a pressure change greater than 5 hPa occurs within one hour, the balloon is considered to be undergoing an altitude control manoeuvre". Could you provide an illustration of either pressure or balloon altitude timeseries that shows such manoeuvre, and clearly displays which part of the dataset is discarded?

**The author team does not see that this has value as a new Figure within the paper. However, we show one example in Figure 2 at the end of this document which shows the excluded region of measurements in red and the data used in analysis around this transition in black. We hope this reassures the reviewer that the methodology used is robust.**

3. Sensor precisions: observations performed on Google balloons were likely not primary intended to provide scientific-class measurements, and stated sensor precisions (p4 l28) are rather large compared with current state-of-the-art meteorological measurements. This is not an issue in itself provided that the impacts of the fairly large measurement uncertainties are precisely assessed. This aspect needs to be improved in the current manuscript:

• First, if one assumes that the uncertainty on the GPS horizontal position is 10 m

(as mentioned in p4, l28), and furthermore that the uncertainties on the two positions separated by t that serve to compute the winds are independent (which is not explicitly stated), then the uncertainty on the derived wind should be 10 p 2=t = 0.23m/s, with t = 1 min (p26, l4). It is only when p 2 is replaced by 2 than one comes to the 0.33m/s value reported in the paper, which I do not understand.

**We thank the reviewer for spotting this error. We have replaced this number in the updated document.**

 c The pressure measurement is used as the vertical coordinate in the interpolation of the reanalysis product onto the balloon position. As stated in the paper (p4., l28 ôĂĂ̌ 31), a 1:5 hPa uncertainy in these measurements is "rather large" and "could potentially lead to uncertainties when vertically interpolating the reanalysis data sets to the balloon locations". While likely true, this sentence stays very qualitative. It would be much helpful if a typical vertical wind shear could be assumed so as to infer a resulting uncertainty on the interpolated wind. When all these measurement/interpolation uncertainties are properly taken into account, one can better know which part of the differences between the balloon observation and analysis is due to the observations or to defficiencies in the analysis (Section 3.1, and Figures 3 and 4).

**A back of the envelope calculation using the hydrostatic equation shows that a 1.5 hPa uncertainty equates to about 300m in altitude. Given a 3.0m/s change over 2km at the bottom of the stratospheric jet in the Southern hemisphere winter (approximated from ERA-Interim climatology) this equates to about 0.4m/s at worst case. This information has therefore been added into the updated document. However, the reality is that this uncertainty would only impact the spread and the findings in this paper are comparable to previous studies. The structure of the errors in Figure 2 also suggests that it is not consistent biases that**

make up the differences observed - more a lack of high frequency detail in the reanalyses.

**We have added the following text into the updated manuscript: "Using the hydrostatic equation shows that a 1.5 hPa pressure uncertainty equates to about 300 m in altitude. Given a 3.0 m/s change over 2 km at the bottom of the stratospheric jet in the Southern hemisphere winter (approximated from ERA-Interim climatology) this equates to about 0.4 m/s at worst case."**

4. Could you also provide confidence intervals in Figure 4, and state which values are significant in Table 2? And please provide only significant digits in this table.

**We have recreated Figure 4 and added the 99% confidence interval around the ERA-Interim mean (black dotted line in Figure 3 at the end of this document). Examination of this figure shows that the confidence interval is similar to the width of the line representing the mean value. The confidence intervals on the standard deviations are also similarly small. Thus, rather than add these lines into the final version of the document. We have added the following text.**

**"Note that the 99% confidence interval associated with the biases is such that they are similar to the width of the line representing the bias."**

**With reference to Table 2, we have calculated the significance linked to the difference in the means of the Loon observations and the reanalysis output using the student's t test and the f test for the significance level for the differences in the variances of the distributions. In every case, the differences between the Loon observations and the reanalysis output are significantly different at greater than the 99% level, largely because of the very large number of data points analysed. We have therefore added the following text into a revised manuscript:**

[Figure]

**"However, the statistical significance linked to the difference in the means of the Loon observations and the reanalysis output have been calculated using the student's t test and the f test for the significance level for the differences in the variances of the distributions. In every case, the differences between the Loon observations and the reanalysis output are significantly different at greater than the 99% level."**

5. One way to identify the uncertainty on the wind measurements is to compute the spectrum of wind disturbances and look where the spectrum becomes flat at high frequencies. The raw timeseries could then even be filtered to eliminate the high-frequency noise, and comparisons with the reanalyses could be made with these filtered timeseries, which would more accurately estimate the analysis defficiciencies. I therefore think that providing the wind spectrum would be a very valuable addition to the article.

**The spectral form and the make-up of the errors is the subject of ongoing work to be developed into another paper. Thus, the author team feel that this suggestion, while of interest, is outside the scope of the current work.**

6. As mentioned on p5, l3 -5, the temperature measurements provided by Loon balloons are those of the "lift gas", and not of the ambient air. In Section 3.3, the authors use an empirical method to correct the lift-gas temperatures from observed diurnal variations that are implicitly assumed to be spurious, and claim that "this [method] is commonly used to correct balloon based temperature measurements". While it is true that such method has been previously used (articles cited in the paper), it was solely used to correct air temperature observations in the lower stratosphere. Its use to correct lift gas temperature measurements, as done here, is more questionnable: one assumption of this method is indeed that the undelying 'true' temperature is not exhibiting diurnal variations (or that the diurnal cycle is less than the sensor uncertainty).

**No, the method does not assume that the 'true' temperatures do not exhibit a diurnal variation. Quite the contrary. It simply assumes that the bias between the true measurements (which show a diurnal variation) and the lift gas temperature measurements (which also show a diurnal variation) is a function of the solar zenith angle only. The robustness of the statistical model that describes these differences demonstrates that this is a valid assumption.**

I doubt that this is the case for the lift gas temperature: the balloon envelop certainly absorbs to some extent the sun radiations, which would unavoidably lead to an increase of the gas temperature during day. It is certainly true that the temperature sensor itself absorbs these radiations, and thus overestimates the gas temperature diurnal cycle. But correcting the measurements to fully eliminate the diurnal cycle is likely excessive.

**We do not correct the measurements to eliminate the diurnal cycle.**

I would thus recommend to discard using the temperature correction, but I would keep Figure 8, and slightly rephrase the sentence on page 9 l 20: it is not only the "quality of the Loon temperature data" which is an issue, it is also the fact that they only measure the gas temperature, which can be quite different to that of the air.

**Yes, and this difference is what we correct for. We believe that the reviewer has not correctly understood the process we have applied for correcting the removing the bias between the lift gas temperatures and the ambient air temperatures.**

[Figure]

Minor points

• p2 l35 to p4 l9 is a long report on previous studies that used similar methodology than the one used in this study. I do not discuss the interest of mentioning these various studies to motivate the present work. I nevertheless think that the discussion could be somewhat synthetized and maybe re-organized by e.g. Earth regions (Northern Hemisphere high latitudes, tropics, Southern hemisphere mid/high latitudes), so as to ease the reader to get a clear picture of these previous results.

**We have rearranged the ordering of this section so that the Northern hemisphere, the equatorial region and then the Southern hemisphere are discussed. But, have made no other changes apart from ordering of paragraphs given that the other reviewer does not raise this criticism.**

• end of introduction: could you provide the plan of your study here?

**The author team do not see the purpose of creating a 'map' of the document and given that this is a stylistic point we have decided not to take-up this suggestion.**

• p6, l10: could you be more specific on the studies that attribute differences to inertia-gravity waves? Could you furthermore state at which latitude the timeseries displayed on Figure 2 were obtained? The frequency of inertia-gravity wave depends on latitude, and it may be worth testing that the apparent period of the short timescale disturbances in the wind timeseries indeed corresponds to the inertial period.

**The attribution of the errors is the subject of ongoing work to be developed into**

another paper. Thus, the author team feel that this suggestion, while of interest, is outside the scope of the current work. However, we have identified the range of latitudes related to the the time series in Figure 2 and added text to that effect in the updated document, specifically:

'The wind observations displayed in Figure 2 were collected between 31 and 48oS and cross the international date line.'

• Sentences on p6 l18 and p7 l3 do not seem to be consistent: does ERA-Interim performs better than MERRA?

We think this confusion stems from us stating that MERRA has the largest mean wind difference, then claiming that it is one of the better performing ones. However, this issue is specifically addressed in the discussion of figure 4. In particular, we identify that the other reanalyses have a similar magnitude of wind bias, but the average over latitude nearly cancels.
* * *
**(a)**

**(b)**

[Figure]

**(c)**

**(d)**

**Fig. 1.** Updated Figure 1

[Figure]

**Fig. 2.** Example of exclusion region after a control period.

[Figure]

[Figure]

**Fig. 3.** Figure 4 including a confidence interval on ERA-Interim bias. Note that black dotted line is similar to width of the green line representing ERA-Interim

---

## Author Comment (AC2) · 16 Sep 2016

**We would like to thank the reviewers for their insightful questions and suggestions for improvement that they have provided. In the remainder of this document bold text identifies responses to reviewers comments.**

**Author team.**

**Reviewer 2 Comments and Responses**

The authors compared zonal and meridional winds of ERA-Interim, MERRA, MERRA–2 and NCEP CFSv2 reanalyses with those derived from X Project Loon. The goal of this study is to test the potential for the Loon to be used in the evaluation of reanalyses. Because the Loon data is not assimilated in each reanalysis, the Loon should provide a useful independent test of atmospheric reanalysis winds. The purpose of this study is well written. This paper includes very new results contributing to reanalysis communities. X Project Loon might have large potential to improve reanalyses in the future. I believe this paper is suitable for the publication in ACP. I have a few comments written below.

<Specific comments> (1) Abstract: "All reanalysis data sets accurately describe the winds, with biases in zonal winds of less than 0.37m/s and meridional biases of less than 0.08m/s. The standard deviation on the differences between Loon and reanalyses zonal winds is latitude dependent, ranging between 2.5 and 3.5 m/s increasing equatorward"

P3L2–3: "Both analyses accurately represented the winds with biases of less than 0.3 m/s and standard deviations ranging from 2.3 to 2.7 m/s", and also P3L8, P3L14, P3L18, P4L3–5 etc.

The bias and standard deviation depend on time scales analyzed (i.e., hourly, daily, monthly). The authors should indicate output intervals analyzed in each case. In addition, as the authors sometimes speculate differences due to gravity waves, differences between reanalyses and observations also depend much on their vertical resolutions. More careful explanations are needed.

Throughout this study our analysis uses this 1 minute temporal resolution data for comparison with interpolated reanalysis data or trajectories derived from that data.

[Figure]

We have now added information on the temporal resolution of the previous studies and also highlighted that the current measurements are compared to the interpolated reanalyses output at a 1 minute interval. The large number of samples this allows in this study is therefore likely to explain the strong statistical significance of our results. For example, we have added the following line in the methodology section of an updated document:

Throughout this study our analysis uses this 1 minute temporal resolution data for comparison with interpolated reanalysis data or trajectories derived from that data.

(2) Kawatani et al. (2016) may be useful for this paper, which shows the standard deviations calculated by monthly mean data among several reanalyses show large values in the equatorial stratosphere, and their geographical distributions in the lower stratosphere are closely related to the density of in-situ radiosonde observations. Kawatani, Y., Hamilton, K., Miyazaki, K., Fujiwara, M., and Anstey, J. A.: Representation of the tropical stratospheric zonal wind in global atmospheric reanalyses, Atmos. Chem. Phys., 16, 6681-6699, doi:10.5194/acp-16-6681-2016, 2016.

We have added the following text in an updated document at the end of the Introduction section:

"More recent work detailed in Kawatani et al. (2016) also suggests that at 50-70a hPa the geographical distributions of the disagreement between the different reanalyses is closely related to the density of radiosonde observations."

(3) P4L21: The authors mention the balloons is considered to be undergoing an

altitude control whenever a pressure change greater than 5 hPa occurs within one hour. Providing height information of Loon flights in Fig. 1 should be useful for reads to Imagine.
We have now included the pressure histogram for all the flights in Figure 1 as requested by Reviewer 1 and we feel that this inclusion covers this point. Please see updated Figure 1 in response to reviewer 1

(4) P4L29–31: "The upper bound on the pressure sensor uncertainty is rather large and could potentially lead to uncertainties when vertically interpolating the reanalyses data sets to the balloon locations" I guess uncertainties depend much on the wind profiles as the vertical resolutions of reanalyses are coarse. Could you specify uncertainties more quantitatively?

Please see the back of the envelope calculation completed for Reviewer 1. We have now added the following text into the updated manuscript:

"Using the hydrostatic equation shows that a 1.5 hPa pressure uncertainty equates to about 300 m in altitude. Given a 3.0 m/s change over 2 km at the bottom of the stratospheric jet in the Southern hemisphere winter (approximated from ERA-Interim climatology) this equates to about 0.4 m/s at worst case."

(5) Table 1: As the top boundary of each reanalysis is different, showing layer numbers between 30 and 70hPa (i.e., the varying pressure levels of the balloon flights) should be useful.

This information has been added to the Table as requested.

**(6) P6L31–33: How the phases of the Quasi-Biennial Oscillation play the role of difference between this study and Podglajen et al. (2014)? The authors could provide their speculation in more details.**
**This point was related to wave propagation differences into the tropical lower stratosphere due to the Holton-Tan effect. However, clearly we do not have enough data to test this and thus have decided to leave out this extra speculation.**

**Interactive comment on Atmos. Chem. Phys. Discuss., doi:10.5194/acp-2016-396, 2016.**

---

## Referee Report (RR1)

Review of
"A comparison of Loon balloon observations and stratospheric reanalysis products"
by Leon S. Friedrich *et al.*

This paper is a revised submission of the article by Friedrich et al. that deals with comparisons between reanalyzed winds and wind observations gathered during Google Loon long-duration balloon flights. The paper also compares balloon trajectories with those computed with reanalyzed winds. I re-iterate my initial statement on the previous version that this study uses an unique dataset to provide very helpful information on reanalysis accuracies, which are widely used to study transport in the atmosphere. I have furthermore found that the authors have thoughtfully addressed most of my comments on the previous version, as well as those of the other reviewer. I therefore recommend publication of this article. I have nevertheless listed a few minor points that the authors may want to address before publication.

**Minor points**

- p7, l146 (end of introduction): I still find that the transition between the end of the introduction, which discusses previous studies, and Section 2 is a somewhat abrupt. I would encourage the authors to either include a short outline of their studies, or a transition sentence.

- p7, l160: remove comma after "while"

- p11,l272 to p12, l283: One likely reason that could explain the differences between this study and Podglajen et al. (2014) regarding reanalysis accuracies in the tropics is that Podglajen et al. (2014) deal with deep tropical balloon flights (within $10°$ of the equator), while this study considers observations mostly southward of $15°$S. It is expected from simple balance argument that mass information provided by spaceborne instruments provides less and less constraints on the wind field as one gets closer to the equator. This is for instance illustrated in Baker et al. (2014) (their Figure 2 notably), which shows that the largest wind errors between models are actually located in the deep tropics (and above oceans).

- p13, l317-324: Another possible reason to explain this difference in trajectory separation is that the balloon flights considered in Hertzog et al. (2004) took place in the stratospheric polar vortex. The separation between the real and simulated balloons in this study was therefore somehow limited by the polar vortex size.

- p16, l388: I agree with the authors' statements on my previous comment. Yet, I find that some of the words used around here are inducing some confusion. For instance, the authors use "SZA bias" while they are referring to differences between the temperature of the lifting gas (i.e. in the balloon envelop) and that of the ambient air. On the next sentence, they carry on with "the solar heating on the lift gas temperature is much more significant than the usual solar bias", i.e. the one which takes place when one is measuring the ambient air temperature. I would suggest to make a distinction between:

    1. a real measurement bias associated with the daytime radiative heating of temperature sensors (used either to measure ambient air or lift gas temperatures). This bias is mostly dependent on the sensor size, coating and ventilation, and is quite unlikely to explain the 30-K difference between the lift gas temperature measurement and the real ambient air temperature,

    2. a physical issue, which is that the daytime lift gas temperature is indeed warmer (by a few tens of degrees) than the ambient air temperature. This issue arises as the balloon envelop absorbs in the UV-visible, and thus conductively heats the lift gas.

    My impression is therefore that using Google Loon temperature measurements to infer air temperature is mostly spoiled by this second item than by the sensor bias itself, which was what was

corrected for in the previous studies mentioned in the article. I would thus rather use "temperature difference" or anything that clearly identify this contrast. I furthermore notice that this second item likely also explains the nighttime lift-gas temperature that are sometimes significantly colder than the ambient air. Here, it is the envelop absorption in the infrared which plays a role: if the balloon flies over high, optically-thick clouds the envelop cools and cools down the lift gas with respect to the ambient air.

**Bibliography**

Baker, W. E., et al., 2014: Lidar-measured wind profiles: the missing link in the Global Oserving System. *Bull. Am. Meteorol. Soc.*, **95**, doi:10.1175/BAMS-D-12-00164.1.

---

## Author Response (AR2)

**Reponses to the comments by Referee 1 are identified below. Note that given that these are 'technical corrections' in the Co-Editors opinion (also see below), we have not gone into great detail of the corrections, but just identified the replacement or additional text where appropriate. Responses to reveiwers comments in bold. Note that we have also added further acknowledgements and corrected some citation errors relative to the last version.**

**Co-Editor Decision: Publish subject to technical corrections** (13 Dec 2016) by Peter Haynes
Comments to the Author:
Both referees recommend that the current version of the paper be published 'as is'. Therefore I am pleased to accept the paper for publication in ACP.

Referee 1 notes a number of further points of clarification that might be made -- some of which seem sensible to me. Therefore please can you consider addressing these prior to providing a final version of the paper. (I am regarding these as possible 'technical corrections' -- the next version of the paper that you provide will simply proceed to publication.)

This paper is a revised submission of the article by Friedrich et al. that deals with comparisons between reanalyzed winds and wind observations gathered during Google Loon long-duration balloon flights. The paper also compares balloon trajectories with those computed with reanalyzed winds. I re-iterate my initial statement on the previous version that this study uses an unique dataset to provide very helpful information on reanalysis accuracies, which are widely used to study transport in the atmosphere. I have furthermore found that the authors have thoughtfully addressed most of my comments on the previous version, as well as those of the other reviewer. I therefore recommend publication of this article. I have nevertheless listed a few minor points that the authors may want to address before publication.

Minor points

- p7, l146 (end of introduction): I still find that the transition between the end of the introduction,
which discusses previous studies, and Section 2 is a somewhat abrupt. I would encourage the
authors to either include a short outline of their studies, or a transition sentence.

**We have now added the following at the end of the introduction:**

**The remainder of this paper documents the Loon observations 5 (Section 2.1), introduces the methodology used in our analysis and specifically details the trajectory model used**

**(Section 2.2). Comparison of the Loon zonal and meridional wind speeds with reanalysis products is then detailed (Section 3.1) and the Loon flight paths are used to examine the accuracy of trajectories derived from the reanalyses in Section 3.2.**

- p7, l160: remove comma after "while"

**Sentence changed to:**

**While super-pressure balloons typically move along isopycnic (constant density) surfaces during the rare occasions of altitude control this is no longer the case.**

- p11,l272 to p12, l283: One likely reason that could explain the differences between this study and Podglajen et al. (2014) regarding reanalysis accuracies in the tropics is that Podglajen et al. (2014) deal with deep tropical balloon flights (within 10◦ of the equator), while this study considers observations mostly southward of 15◦S. It is expected from simple balance argument that mass information provided by spaceborne instruments provides less and less constraints on the wind field as one gets closer to the equator. This is for instance illustrated in Baker et al. (2014) (their Figure 2 notably), which shows that the largest wind errors between models are actually located in the deep tropics (and above oceans).

**Added the following sentence:**

**The fact that Podglajen et al. (2014) also examine a narrower latitude band (within 10⁰ of the equator) may also be important.**

- p13, l317-324: Another possible reason to explain this difference in trajectory separation is that the balloon flights considered in Hertzog et al. (2004) took place in the stratospheric polar vortex. The separation between the real and simulated balloons in this study was therefore somehow limited by the polar vortex size.

**After examination of Figure 1 in Hertzog et al. (2004) we are not sure that this point is significant and have therefore not added any text.**

- p16, l388: I agree with the authors' statements on my previous comment. Yet, I find that some of the words used around here are inducing some confusion. For instance, the authors use "SZA bias" while they are referring to differences between the temperature of the lifting gas (i.e. in the balloon envelop) and that of the ambient air. On the next sentence, they carry on with "the solar heating on the lift gas temperature is much more significant than the usual solar bias", i.e. the one which takes place when one is measuring the ambient air temperature. I would suggest to make a distinction between:

1. a real measurement bias associated with the daytime radiative heating of temperature sensors (used either to measure ambient air or lift gas temperatures). This bias is mostly dependent on the sensor size, coating and ventilation, and is quite unlikely to explain the 30-K difference between the lift gas temperature measurement and the real ambient air temperature,
2. a physical issue, which is that the daytime lift gas temperature is indeed warmer (by a few tens of degrees) than the ambient air temperature. This issue arises as the balloon envelop absorbs in the UV-visible, and thus conductively heats the lift gas.

My impression is therefore that using Google Loon temperature measurements to infer air temperature is mostly spoiled by this second item than by the sensor bias itself, which was what was corrected for in the previous studies mentioned in the article. I would thus rather use "temperature difference" or anything that clearly identify this contrast. I furthermore notice that this second item likely also explains the nighttime lift-gas temperature that are sometimes significantly colder than the ambient air. Here, it is the envelop absorption in the infrared which plays a role: if the balloon flies over high, optically-thick clouds the envelop cools and cools down the lift gas with respect to the ambient air.

**We have rewritten the section to replace SZA bias with differences. Updated text below:**

**There are several difficulties associated with the Loon temperature data. As previously stated, the data result from measurements of the lift gas temperature and not of the ambient air, resulting in strong solar zenith angle (SZA) dependent differences between the lift gas temperature and the ambient air temperature. These may result from the combination of the daytime radiative heating of temperature sensors and, we speculate, the balloon envelop absorbing in the UV-visible range. Additionally, although we are not aware of the specific instruments used, it seems that the thermometer used has a high uncertainty and is intended as a diagnostic instrument rather than for scientific data collection. An example of balloon–reanalysis temperature differences is show in Fig. 8. The temperature differences between the lift-gas and ambient air can be corrected through the use of a correction function, as is commonly done to adjust for temperature measurement biases arising due to radiative heating of the temperature sensors (Hertzog et al., 2004, 5 2006; Knudsen et al., 2006), but it should be noted that the impact of solar heating on the lift gas temperature is much more significant than the usual solar bias, up to +30 K as opposed to the typical $\sim 1.5$ K. The temperature differences can be modelled as:**

$$T_{diff} = \alpha + \begin{cases} \beta(1 - e^{(\theta-95)/\lambda_0}) + \gamma e^{-(\theta-90)^2/\lambda_1} & \theta \le 90 \\ \beta(1 - e^{(\theta-95)/\lambda_0}) + \gamma e^{-(\theta-90)^2/\lambda_2} & 90 < \theta \le 95 \\ \gamma e^{-(\theta-90)^2/\lambda_2} & 95 < \theta \le 150 \\ \gamma e^{-(\theta-90)^2/\lambda_2} + \delta \cdot (\theta - 150) & 150 < \theta \end{cases}$$

where $\alpha$, $\beta$, $\gamma$, $\delta$, $\lambda_0$, $\lambda_1$ and $\lambda_2$ are fit coefficients determined from a linear least squares regression. After removing some flights with anomalous observations (unreasonably large differences, questionable GPS or pressure data), we use temperature data from every second flight to fit the correction function, and then apply this correction to the remaining flights. The fitted parameters are provided in Table 4, and Figure 9 shows the CFSv2 temperature differences with and without the correction applied. Application of the correction functions reduces the mean Loon-reanalyses temperature differences to a few degrees, significantly improving the utility of the Loon temperature measurements, however the standard deviation and the shorter term, day–to–day differences are still much greater than observed in other studies.

Ignoring the differences between lift-gas and ambient temperatures by focusing only on the nighttime measurements, we still find a standard deviations of ∼ 6 K while other balloon studies typically have biases and standard deviations less than 2 K. Additionally the nighttime measurements show interesting behaviour with common consistent night-long differences of up to ±10 K. Considering the upper bound on the thermometer uncertainty provided by the Loon team, the significant difference which is much greater than those usually dealt with using correction functions, and the unusually inaccurate night-time temperatures, leads us to conclude that currently the quality of the Loon temperature data means it is of little value in assessing the quality of the reanalyses.